META-RESEARCH

# International authorship and collaboration across bioRxiv preprints

**Abstract** Preprints are becoming well established in the life sciences, but relatively little is known about the demographics of the researchers who post preprints and those who do not, or about the collaborations between preprint authors. Here, based on an analysis of 67,885 preprints posted on bioRxiv, we find that some countries, notably the United States and the United Kingdom, are overrepresented on bioRxiv relative to their overall scientific output, while other countries (including China, Russia, and Turkey) show lower levels of bioRxiv adoption. We also describe a set of 'contributor countries' (including Uganda, Croatia and Thailand): researchers from these countries appear almost exclusively as non-senior authors on international collaborations. Lastly, we find multiple journals that publish a disproportionate number of preprints from some countries, a dynamic that almost always benefits manuscripts from the US.

**RICHARD J ABDILL, ELIZABETH M ADAMOWICZ AND RAN BLEKHMAN\***

**\*For correspondence:** blekhman@umn.edu

## Introduction

Preprints are being shared at an unprecedented rate in the life sciences (*Narock and Goldstein, 2019*; *Abdill and Blekhman, 2019b*): since 2013, more than 90,000 preprints have been posted to bioRxiv.org, the largest preprint server in the field, including a total of 29,178 in 2019 alone (*Abdill and Blekhman, 2019a*). In addition to allowing researchers to share their work independently of publication at a traditional journal, there is evidence that published papers receive more citations if they first appeared as preprints (*Fu and Hughey, 2019*; *Fraser et al., 2020*). Some journals also search preprint servers to solicit submissions (*Barsh et al., 2016*; *Vence, 2017*), and there are various initiatives to encourage and facilitate the peer review of preprints, such as In Review (https://www.researchsquare.com/publishers/in-review), Review Commons (https://www.review-commons.org), and Preprint Review (*eLife, 2020*). However, relatively little is known about who is benefiting from the growth of preprints or how this new approach to publishing is affecting different populations of researchers (*Penfold and Polka, 2020*).

Academic publishing has grappled for decades with hard-to-quantify concerns about factors of success that are not directly linked to research quality. Studies have found bias in favor of wealthy, English-speaking countries in citation count (*Akre et al., 2011*) and editorial decisions (*Nuñez et al., 2019*; *Saposnik et al., 2014*; *Okike et al., 2008*; *Ross et al., 2006*), and there have long been concerns regarding how peer review is influenced by factors such as institutional prestige (*Lee et al., 2013*). Preprints have been praised as a democratizing influence on scientific communication (*Berg et al., 2016*), but a critical question remains: where do they come from? More specifically, which countries are participating in the preprint ecosystem, how are they working with each other, and what happens when they do? Here, we aim to answer these questions by analyzing a dataset of all preprints posted to bioRxiv between its launch in 2013 and the end of 2019. After collecting author-level metadata for each preprint, we used each

**Figure 1.** Preprints per country. (a) A heat map indicating the number of preprints per country, based on the institutional affiliation of the senior author. The color coding uses a log scale. (b) The total preprints attributed to the seven most prolific countries. The x-axis indicates total preprints listing a senior author from a country; the y-axis indicates the country. The 'Other' category includes preprints from all countries not listed in the plot. (c) Similar to panel b, but showing the total preprints listing at least one author from the country in any position, not just the senior position. (d) Proportion

*Figure 1 continued on next page*

*Figure 1 continued*

of total senior-author preprints from each country (y-axis) over time (x-axis), starting in November 2013 and continuing through December 2019. Each colored segment indicates the proportion of total preprints attributed to a single country (using same color scheme as panels (**b** and **c**), as of the end of the month indicated on the x-axis.

The online version of this article includes the following source data and figure supplement(s) for figure 1:

**Source data 1.** Preprints per country.
**Source data 2.** Preprint counting methods at the country level.
**Figure supplement 1.** Preprint-level collaboration.
**Figure supplement 2.** Preprints with no country assignment.

author's institutional affiliation to summarize country-level participation and outcomes.

## Results

### Country level bioRxiv participation over time

We retrieved author data for 67,885 preprints for which the most recent version was posted before 2020. First, we attributed each preprint to a single country, using the affiliation of the last individual in the author list, considered by convention in the life sciences to be the 'senior author' who supervised the work (see **Methods**). North America, Europe and Australia dominate the top spots (*Figure 1a*): 26,598 manuscripts (39.2%) have a last author from the United States (US), followed by 7151 manuscripts (10.5%) from the United Kingdom (UK; *Figure 1b*), though China (4.1%), Japan (1.9%) and India (1.8%) are the sources of more than

1200 preprints each (*Table 1*). Brazil, with 704 manuscripts, has the 15th-most preprints and is the first South American country on the list, followed by Argentina (163 preprints) in 32nd place. South Africa (182 preprints) is the first African country on the list, in 29th place, followed by Ethiopia (57 preprints) in 42nd place (*Figure 1—source data 1*). It is noticeable that South Africa and Ethiopia both have high opt-in rates for a program operated by PLOS journals that enabled submissions to be sent directly to bioRxiv (*PLOS, 2019*). We found similar results when we looked at which countries were most highly represented based on authorship at any position (*Table 1*). Overall, US authors appear on the most bioRxiv preprints – 34,676 manuscripts (51.1%) include at least one US author (*Figure 1c*).

Over time, the country-level proportions on bioRxiv have remained remarkably stable (*Figure 1d*), even as the number of preprints

**Table 1.** Preprints per country.

| Country | Preprints, senior author (proportion) | Preprints, any author (proportion) |
| --- | --- | --- |
| United States | 26,598 (39.2%) | 34,676 (51.1%) |
| United Kingdom | 7151 (10.5%) | 11,578 (17.1%) |
| (Unknown) | 4985 (7.3%) | 17,635 (26.0%) |
| Germany | 3668 (7.3%) | 7157 (10.5%) |
| France | 2863 (4.2%) | 5218 (7.7%) |
| China | 2778 (4.1%) | 4609 (6.8%) |
| Canada | 2380 (3.5%) | 4409 (6.5%) |
| Australia | 1755 (2.6%) | 3260 (4.8%) |
| Switzerland | 1364 (2.0%) | 2779 (4.1%) |
| Netherlands | 1291 (1.9%) | 2764 (4.1%) |
| Japan | 1263 (1.9%) | 2287 (3.4%) |
| India | 1212 (1.8%) | 1769 (2.6%) |

All 11 countries with more than 1000 preprints attributed to a senior author affiliated with that country. The percentages in the 'Preprints, any author' column sum to more than 100% because preprints may be counted for more than one country. A full list of countries is provided in **Figure 1—source data 1**.

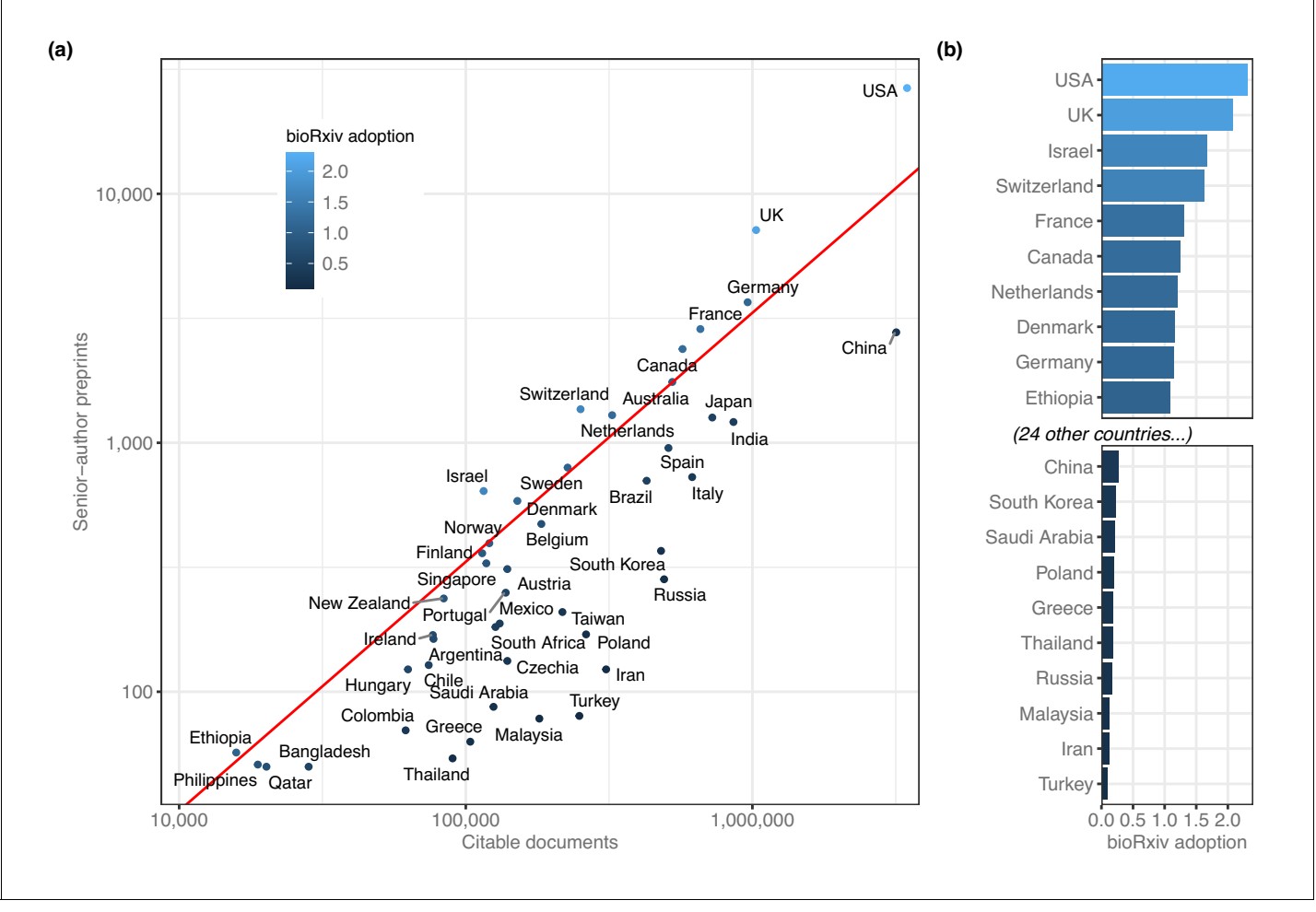

**Figure 2.** BioRxiv adoption per country. (a) Correlation between two scientific output metrics. Each point is a country; the x-axis (log scale) indicates the total citable documents attributed to that country from 2014 to 2019, and the y-axis (also log scale) indicates total senior-author preprints attributed to that country overall. The red line demarcates a 'bioRxiv adoption' score of 1.0, which indicates that a country's share of bioRxiv preprints is identical to its share of general scholarly outputs. Countries to the left of this line have a bioRxiv adoption score greater than 1.0. A score of 2.0 would indicate that its share of preprints is twice as high as its share of other scholarly outputs (See **Discussion** for more about this measurement.) (b) The countries with the 10 highest and 10 lowest bioRxiv adoption scores. The x-axis indicates each country's adoption score, and the y-axis lists each country in order. All panels include only countries with at least 50 preprints.

The online version of this article includes the following source data for figure 2:

**Source data 1.** Country productivity and bioRxiv adoption.

grew exponentially. For example, at the end of 2015 Germany accounted for 4.7% of bioRxiv's 2460 manuscripts, and at the end of 2019 it was responsible for 5.4% of 67,885 preprints. However, the proportion of preprints from countries outside the top seven contributing countries is growing slowly (*Figure 1d*): from 19.4% at the end of 2015 to 23.1% at the end of 2019, by which time bioRxiv was hosting preprints from senior authors affiliated with 136 countries.

### Preprint adoption relative to overall scientific output

We noted that some patterns may be obscured by countries that had hundreds or thousands of times as many preprints as other countries, so we re-evaluated these ranks after adjusting for overall scientific output (*Figure 2a*). The corrected measurement, which we call 'bioRxiv adoption,' is the proportion of preprints from each country divided by the proportion of worldwide research outputs from that country (see **Methods**). The US posted 26,598 preprints and published about 3.5 million citable documents,

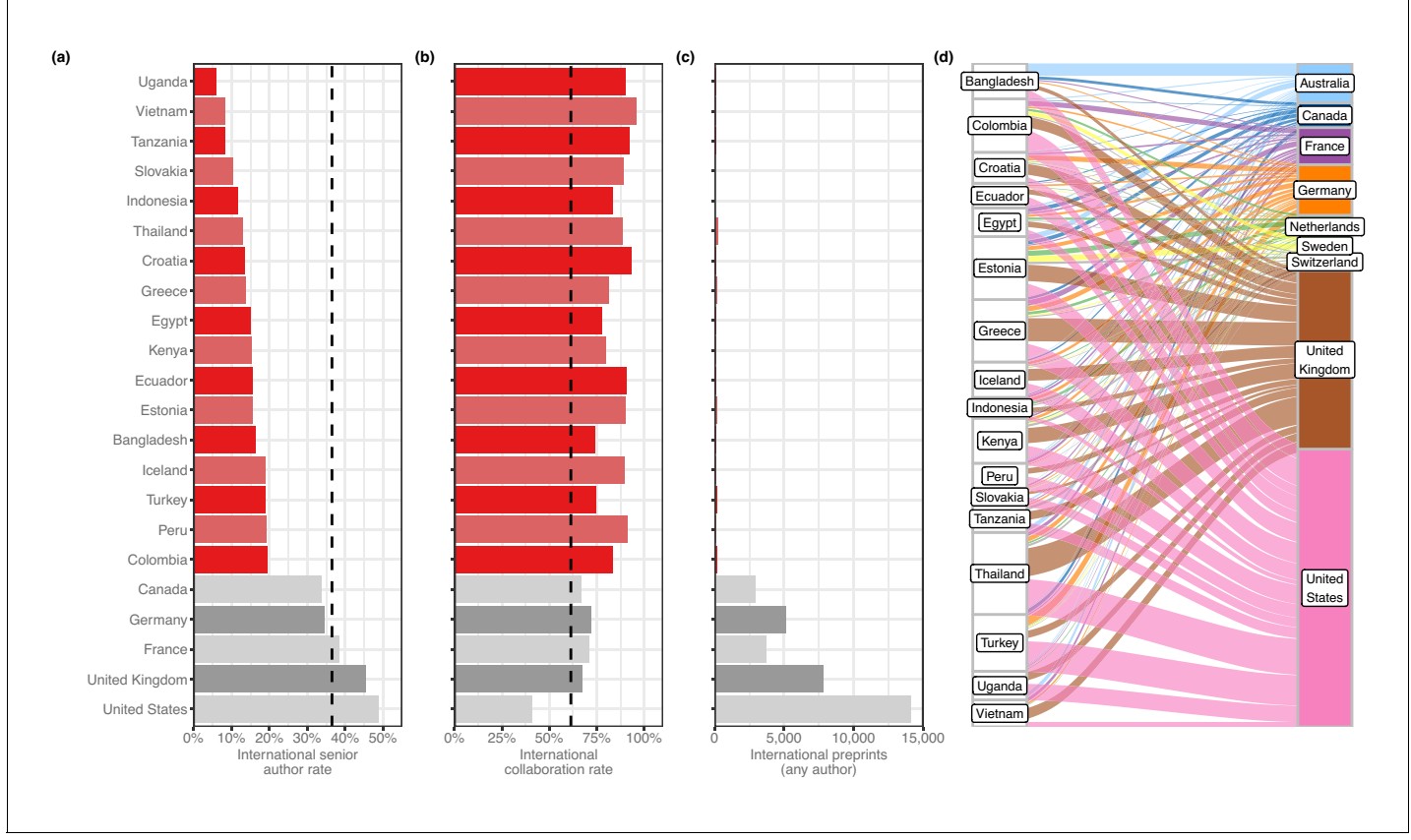

**Figure 3.** Contributor countries. (**a**) Bar plot indicating the international senior author rate (y-axis) by country (x-axis) – that is, of all international preprints with a contributor from that country, the percentage of them that include a senior author from that country. All 17 contributor countries are listed in red, with the five countries with the highest senior-author rates (in grey) for comparison. (**b**) A bar plot with the same y-axis as panel (**a**). The x-axis indicates the international collaboration rate, or the proportion of preprints with a contributor from that country that also include at least one author from another country. (**c**) is a bar plot indicating the total international preprints featuring at least one author from that country (the median value per country is 19). (**d**) On the left are the 17 contributor countries. On the right are the countries that appear in the senior author position of preprints that were co-authored with contributor countries. (Supervising countries with 25 or fewer preprints with contributor countries were excluded from the figure.) The width of the ribbons connecting contributor countries to senior-author countries indicates the number of preprints supervised by the senior-author country that included at least one author from the contributor country. Statistically significant links were found between four combinations of supervising countries and contributors: Australia and Bangladesh (Fisher's exact test, $q = 1.01 \times 10^{-11}$); the UK and Thailand ($q = 9.54 \times 10^{-4}$); the UK and Greece ($q = 6.85 \times 10^{-3}$); and Australia and Vietnam ($q = 0.049$). All p-values reflect multiple-test correction using the Benjamini–Hochberg procedure.

The online version of this article includes the following source data and figure supplement(s) for figure 3:

**Source data 1.** Combinations of senior authors with collaborator countries.

**Source data 2.** Links between contributor countries and the senior-author countries they write with.

**Source data 3.** International collaboration.

**Figure supplement 1.** Map of contributor countries.

**Figure supplement 2.** International collaboration correlations.

**Figure supplement 3.** Correlation between three measurements of international collaboration.

for a bioRxiv adoption score of 2.31 (*Figure 2b*). Nine of the 12 countries with adoption scores above 1.0 were from North America and Europe, but Israel has the third-highest score (1.67) based on its 640 preprints. Ethiopia has the 10th-highest bioRxiv adoption (1.08): though only 57 preprints list a senior author with an affiliation in Ethiopia, the country had a total of

15,820 citable documents published between 2014 and 2019 (*Figure 2—source data 1*).

By comparison, some countries are present on bioRxiv at much lower frequencies than would be expected, given their overall participation in scientific publishing (*Figure 2b*): Turkey published 249,086 citable documents from 2014 through 2019 but was the senior author on only

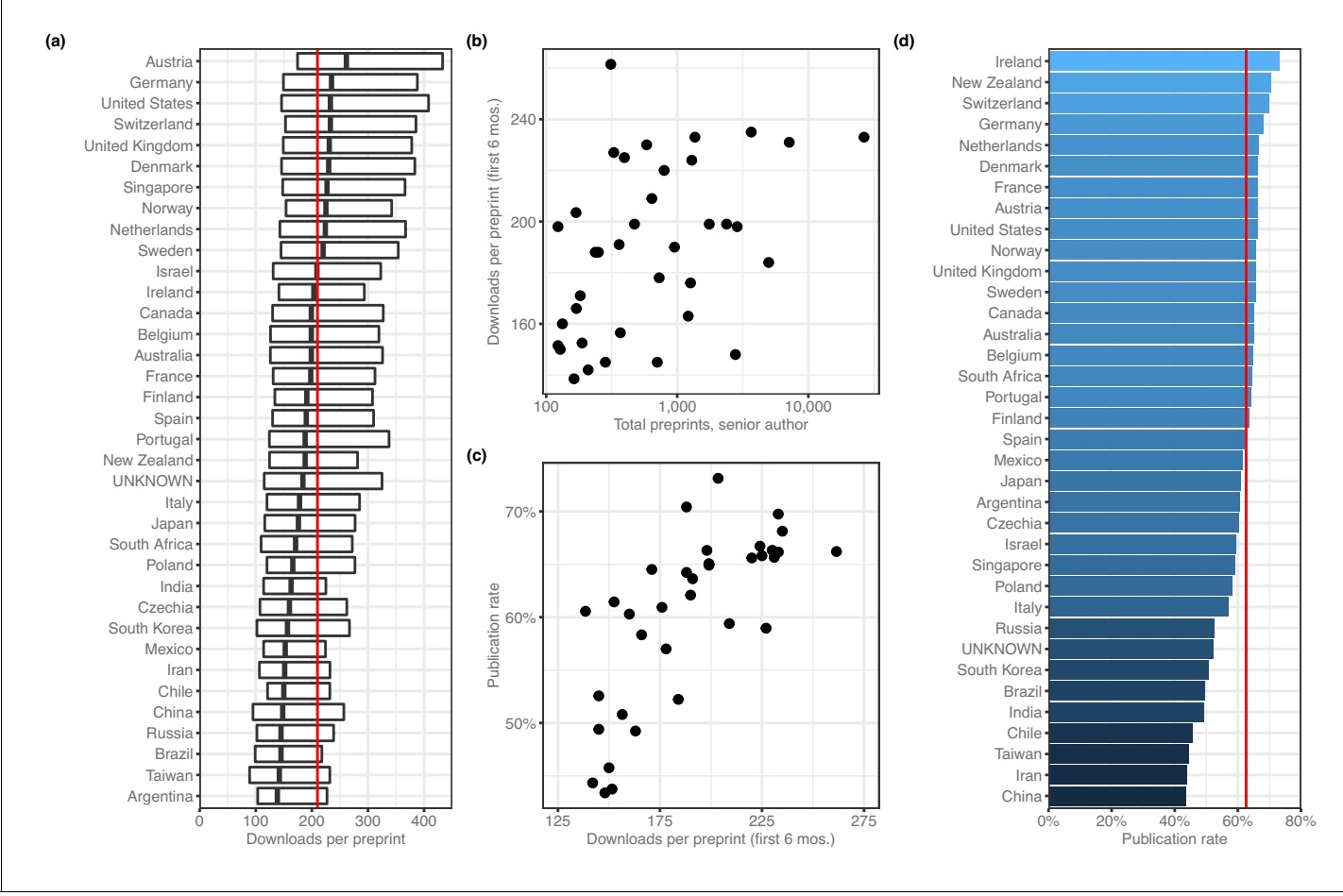

**Figure 4.** Preprint outcomes. All panels include countries with at least 100 senior-author preprints. (a) A box plot indicating the number of downloads per preprint for each country. The dark line in the middle of the box indicates the median, and the ends of each box indicate the first and third quartiles, respectively. 'Whiskers' and outliers were omitted from this plot for clarity. The red line indicates the overall median. (b) A plot showing the relationship (Spearman's $\rho$ = 0.485, p=0.00274) between total preprints and downloads. Each point represents a single country. The x-axis indicates the total number of senior-author preprints attributed to the country. The y-axis indicates the median number of downloads for those preprints. (c) A plot showing the relationship (Spearman's $\rho$ = 0.777, p=2.442 $\times$ 10$^{-8}$) between downloads and publication rate. Each point represents a single country. The x-axis indicates the median number of downloads for all preprints listing a senior author affiliated with that country. The y-axis indicates the proportion of preprints posted before 2019 that have been published. (d) A bar plot indicating the proportion of preprints posted before 2019 that are now flagged as 'published' on the bioRxiv website. The x-axis (and color scale) indicates the proportion, and the y-axis lists each country. The red line indicates the overall publication rate.

The online version of this article includes the following source data for figure 4:

**Source data 1.** Published pre-2019 preprints by country.
**Source data 2.** Publication rates and DOI usage.

80 preprints, for a bioRxiv adoption score of 0.10. Russia (283 preprints), Iran (123 preprints) and Malaysia (78 preprints) all have adoption scores below 0.18. The largest country with a low adoption score is China (3,176,571 citable documents; 2778 preprints; bioRxiv adoption = 0.26), which published more than 15% of the world's citable documents (according to SCImago) but was the source of only 4.1% of preprints (*Figure 2a*).

### Patterns and imbalances in international collaboration

After analyzing preprints using senior authorship, we also evaluated interactions *within* manuscripts to better understand collaborative patterns found on bioRxiv. We found the number of authors per paper increased from 3.08 in 2014 to 4.56 in 2019 (*Figure 1—figure supplement 1*). The monthly average authors per preprint has increased linearly with time (Pearson's

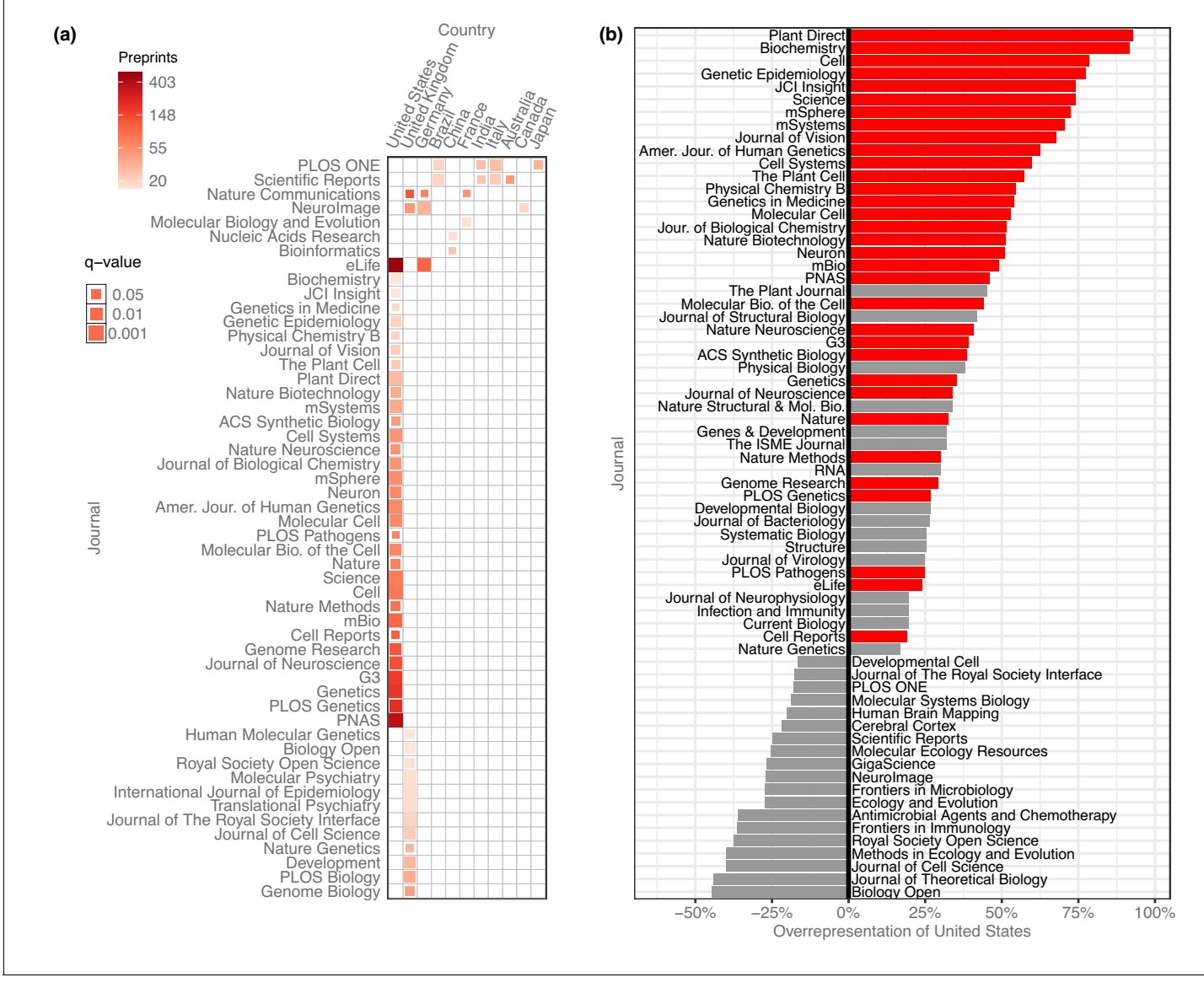

**Figure 5.** Overrepresentation of US preprints. (a) A heat map indicating all disproportionately strong (q < 0.05) links between countries and journals, for journals that have published at least 15 preprints from that country. Columns each represent a single country, and rows each represent a single journal. Colors indicate the raw number of preprints published, and the size of each square indicates the statistical significance of that link—larger squares represent smaller q-values. See *Figure 5—source data 1* for the results of each statistical test. (b) A bar plot indicating the degree to which US preprints are over- or under-represented in a journal's published bioRxiv preprints. The y-axis lists all the journals that published at least 15 preprints with a US senior author. The x-axis indicates the overrepresentation of US preprints compared to the expected number: for example, a value of '0%' would indicate the journal published the same proportion of US preprints as all journals combined. A value of '100%' would indicate the journal published twice as many U. preprints as expected, based on the overall representation of the US among published preprints. Journals for which the difference in representation was less than 15% in either direction are not displayed. The red bars indicate which of these relationships were significant using the Benjamini–Hochberg-adjusted results from $\chi^2$ tests shown in panel A.

The online version of this article includes the following source data for figure 5:

**Source data 1.** Journal–country links.

r = 0.949, p=8.73 × 10$^{-38}$), a pattern that has also been observed, at a less dramatic rate, in published literature (*Adams et al., 2005*; *Wuchty et al., 2007*; *Bordons et al., 2013*). Examining the number of countries represented in each preprint (*Figure 1—figure supplement 1*), we found that 24,927 preprints (36.7%) included authors from two or more countries; 3041 preprints (4.5%) were from four or more countries, and one preprint, 'Fine-mapping of

150 breast cancer risk regions identifies 178 high confidence target genes,' listed 343 authors from 38 countries, the most countries listed on any single preprint. The mean number of countries represented per preprint is 1.612, which has remained fairly stable since 2014 despite steadily growing author lists overall (*Figure 1—figure supplement 1*).

We then looked at countries appearing on at least 50 international preprints to examine basic patterns in international collaboration. We found that a number of countries with comparatively low output contributed almost exclusively to international collaborations: for example, of the 76 preprints that had an author with an affiliation in Vietnam, 73 (96%) had an author from another country. Upon closer examination, we found these countries were part of a larger group, which we call 'contributor countries,' that (1) appear mostly on preprints with authors from other countries, but (2) seldom as the senior author. For this analysis, we defined a contributor country as one that has contributed to at least 50 international preprints but appears in the senior author position of less than 20% of them. (We excluded countries with less than 50 preprints to minimize the effect of dynamics that could be explained by countries with just one or two labs that frequently worked with international collaborators.) 17 countries met these criteria (*Figure 3—figure supplement 1*): for example, of the 84 international preprints that had an author with an affiliation in Uganda, only 5 (6%) had an author from Uganda in the senior author position. This figure was also less than 12% for Vietnam, Tanzania, Slovakia and Indonesia: by comparison, the figure for the US was 48.7% (*Figure 3a*).

In addition to a high percentage of international collaborations and a low percentage of senior-author preprints, another characteristic of contributor countries is a comparatively low number of preprints overall. To define this subset of countries more clearly, we examined whether there was a relationship between any of the three factors we identified across all countries with at least 50 international preprints. We found consistent patterns for all three (see Methods). First, countries with fewer international collaborations also tend to appear as senior author on a smaller proportion of those preprints (*Figure 3—figure supplement 2a*). Second, we also observed a negative correlation between *total* international collaborations and international collaboration *rate* – that is, the proportion of preprints a country contributes to that include at least one contributor from another country (*Figure 3—figure supplement 2b*). This indicates that countries with mostly international preprints (*Figure 3b*) also tended to have *fewer* international collaborations (*Figure 3c*) than other countries. Third, we found a negative correlation between international collaboration rate and the proportion of international preprints for which a country appears as senior author (*Figure 3—figure supplement 2c*), demonstrating that countries that appear mostly on international preprints (*Figure 3b*) are less likely to appear as senior author of those preprints (*Figure 3*). Similar patterns have been observed in previous studies: *González-Alcaide et al., 2017* found countries ranked lower on the Human Development Index participated more frequently in international collaborations, and a review of oncology papers found that researchers from low- and middle-income countries collaborated on randomized control trials, but rarely as senior author (*Wong et al., 2014*).

After generating a list of preprints with authors from contributor countries, we examined which countries appeared most frequently in the senior author position of those preprints (*Figure 3d*). Among the 2133 preprints with an author from a contributor country, 494 (23.2%) had a senior author listing an affiliation in the US (*Figure 3—source data 1*). The UK was listed as senior author on the next-most preprints with contributor countries, at 318 (14.9%), followed by Germany (4.2%) and France (3.1%). Given the large differences in preprint authorship between countries, we tested which of these senior-author relationships was disproportionately large. Using Fisher's exact test (see **Methods**), we found four links between contributor countries and senior-author countries that were significant (*Figure 3—source data 2*). The strongest link is between Bangladesh and Australia: of the 82 international preprints with an author from Bangladesh, 22 list a senior author with an affiliation in Australia. Authors in Vietnam appear with disproportionate frequency on preprints with a senior author in Australia as well (9 of 67 preprints). The other two links are to senior authors in the UK, with contributing authors from Thailand (50 of 187 preprints) and Greece (41 of 155 preprints).

## Differences in preprint downloads and publication rates

After quantifying which countries were posting preprints, we also examined whether there were differences in preprint outcomes between

countries. We obtained monthly download counts for all preprints, as well as publication status, the publishing journal, and date of publication for all preprints flagged as 'published' on bioRxiv (see Methods). We then evaluated country-level patterns for the 36 countries with at least 100 senior-author preprints.

When evaluating downloads per preprint, we used only download numbers from each preprint's first six months online, which would capture the majority of downloads for most preprints (*Abdill and Blekhman, 2019b*) while minimizing the effect of the 'long tail' of downloads that would be longer for countries that were earlier adopters. Using this measurement, the median number of PDF downloads per preprint is 210 (*Figure 4a*). Among countries with at least 100 preprints, Austria has the highest median downloads per preprint, with 261.5, followed by Germany (235.0), Switzerland (233.0) and the US (233.0). Argentina has the lowest median, at 138.5 downloads; next-fewest were Taiwan (142), Brazil (145) and Russia (145). To examine whether these results were influenced by changes in downloads per preprint over time, we re-analyzed the data after dividing each preprint's download count by the median download count of all preprints posted in the same month. The country-level medians of the adjusted downloads per paper are highly correlated (Spearman's rho = 0.989, p=3.33 $\times$ $10^{-109}$) with the unadjusted median downloads per paper, indicating there is no influence from countries posting more preprints during times in which many preprints were downloaded in general. Across all countries with at least 100 preprints, there was a weak correlation between total preprints attributed to a country and the median downloads per preprint (*Figure 4b*), and another correlation between median downloads per preprint and each country's publication rate (*Figure 4c*).

Next, we examined country-level publication rates by assigning preprints posted prior to 2019 to countries using the affiliation of the senior author, then measuring the proportion of those preprints flagged as 'published' on the bioRxiv website. Overall, 62.6% of pre-2019 preprints were published (*Figure 4—source data 1*). Ireland had the highest publication rate (49/67 = 73.1%; *Figure 4d*), followed by New Zealand (100/142; 70.4%) and Switzerland (505/724; 69.8%). China (588/1355, 43.4%) had the lowest publication rate, with Iran and Taiwan also in the bottom three.

## Preprint publication patterns between countries and journals

After evaluating the country-level publication rates, we examined which journals were publishing these preprints and whether there were any meaningful country-level patterns (*Figure 5*). We quantified how many senior-author preprints from each country were published in each journal and used the $\chi^2$ test (with Yates's correction for continuity) to examine whether a journal published a disproportionate number of preprints from a given country, based on how many preprints from that country were published overall. To minimize the effect of journals with differing review times, we limited the analysis to preprints posted before 2019, resulting in a total of 23,102 published preprints.

After controlling the false-discovery rate using the Benjamini–Hochberg procedure, we found 63 significant links between journals and countries, of journal–country links with at least 15 preprints (*Figure 5a*). 11 countries had links to journals that published a disproportionate number of their preprints, but the US had more links than any other country. 33 of the 63 significant links were between a journal and the US: the US is listed as the senior author on 41.7% of published preprints, but accounts for 74.5% of all bioRxiv preprints published in *Cell*, 72.7% of preprints published in *Science*, and 61.0% of those published in *Proceedings of the National Academy of Sciences* (*PNAS*) (*Figure 5b*).

## Discussion

Our study represents the first comprehensive, country-level analysis of bioRxiv preprint publication and outcomes. While previous studies have split up papers into 'USA' and 'everyone else' categories in biology (*Fraser et al., 2020*) and astrophysics (*Schwarz and Kennicutt, 2004*), our results provide a broad picture of worldwide participation in the largest preprint server in biology. We show that the US is by far the most highly represented country by number of preprints, followed distantly by the UK and Germany.

By adjusting preprint counts by each country's overall scientific output, we were able to develop a 'bioRxiv adoption' score (*Figure 2*), which showed the US and the UK are overrepresented while countries such as Turkey, Iran and Malaysia are underrepresented even after accounting for their comparatively low scientific output. Open science advocates have argued that there should not be a 'one size fits all'

approach to preprints and open access (*Humberto and Babini, 2020*; *ALLEA, 2018*; *Becerril-García, 2019*; *Mukunth, 2019*), and further research is required to determine what drives certain countries to preprint servers—what incentives are present for biologists in Finland but not Greece, for example. There is also more to be done regarding the trade-offs of using a more distributed set of repositories that are specific to disciplines or countries (e.g. INA-Rxiv in Indonesia or PaleorXiv for paleontology), which could also influence the observed levels of bioRxiv adoption. Our results make it clear that those reading bioRxiv (or soliciting submissions from the platform) are reviewing a biased sample of worldwide scholarship.

There are two findings that may be particularly informative about the state of open science in biology. First, we present evidence of contributor countries—countries from which authors appear almost exclusively in non-senior roles on preprints led by authors from more prolific countries (*Figure 3*). While there are many reasons these dynamics could arise, it is worth noting that the current corpus of bioRxiv preprints contains the same familiar disparities observed in published literature (*Mammides et al., 2016*; *Burgman et al., 2015*; *Wong et al., 2014*; *González-Alcaide et al., 2017*). Critically, we found the three characteristics of contributor countries (low international collaboration *count*, high international collaboration *rate*, low international senior author rate) are strongly correlated with each other (*Figure 3*). When looking at international collaboration using pairwise combinations of these three measurements, countries fall along tidy gradients (*Figure 3—figure supplement 3*)—which means not only that they can be used to delineate properties of contributor countries, but that if a country fits even one of these criteria, they are more likely to fit the other two as well.

Second, we found numerous country-level differences in preprint outcomes, including a positive correlation at the country level between downloads per preprint and publication rate (*Figure 4c*). This raises an important consideration that when evaluating the role of preprints, some benefits may be realized by authors in some countries more consistently than others. If one of the goals of preprinting one's work is to solicit feedback from the community (*Sarabipour et al., 2019*; *Sever et al., 2019*), what are the implications of the average Brazilian preprint receiving 37% fewer downloads than the average Dutch preprint? Do preprint authors from the most-downloaded countries (mostly in western Europe) have broader social-media reach than authors in low-download countries such as Argentina and Taiwan? What role does language play in outcomes, and why do countries that get more downloads also tend to have higher publication rates? We also found some journals had particularly strong affinities for preprints from some countries over others: even when accounting for differing publication rates across countries, we found dozens of journal–country links that disproportionately favored the US and UK. While it's possible this finding is coincidental, it demonstrates that journals can embrace preprints while still perpetuating some of the imbalances that preprints could be theoretically alleviating.

Our study has several limitations. First, bioRxiv is not the only preprint server hosting biology preprints. For example, arXiv's Quantitative Biology category held 18,024 preprints at the end of 2019 (https://arxiv.org/help/stats/2019_by_area/index), and repositories such as Indonesia's INA-Rxiv (https://osf.io/preprints/inarxiv/) hold multidisciplinary collections of country-specific preprints. We chose to focus on bioRxiv for several reasons: primarily, bioRxiv is the preprint server most broadly integrated into the traditional publishing system (*Barsh et al., 2016*; *Vence, 2017*; *eLife, 2020*). In addition, bioRxiv currently holds the largest collection of biology preprints, with metadata available in a format we were already equipped to ingest (*Abdill and Blekhman, 2019c*). Analyzing data from only a single repository also avoids the issue of different websites holding metadata that is mismatched or collected in different ways. Comparing publication rates between repositories would also be difficult, particularly because bioRxiv is one of the few with an automated method for detecting when a preprint has been published. Second, this 'worldwide' analysis of preprints is explicitly biased toward English-language publishing. BioRxiv accepts submissions only in English, and the primary motivation for this work was the attention being paid to bioRxiv by organizations based mostly in the US and western Europe. In addition, bibliometrics databases such as Scopus and Web of Science have well-documented biases in favor of English-language publications (*Mongeon and Paul-Hus, 2016*; *Archambault et al., 2006*; *de Moya-Anegón et al., 2007*), which could have an effect on observed publication rates and the bioRxiv adoption scores that depend on scientific output derived from Scopus.

There were also 4985 preprints (7.3%) for which we were not able to confidently assign a country of origin. An evaluation of these (see Methods) showed that the most prolific countries were also underrepresented in the 'unknown' category, compared to the 148 other countries with at least one author. While it is impractical to draw country-specific conclusions from this, it suggests that the preprint counts for countries with comparatively low participation may be slightly higher than reported, an issue that may be exacerbated in more granular analyses, such as at the institutional level. Country-level differences in metrics such as downloads and publication rate may also be confounded with field-level differences: on average, genomics preprints are downloaded twice as many times as microbiology preprints (*Abdill and Blekhman, 2019b*), for example, so countries with a disproportionate number of preprints in a particular field could receive more downloads due to choice of topic, rather than country of origin. Further study is required to determine whether these two factors are related and in which direction.

In summary, we find country-level participation on bioRxiv differs significantly from existing patterns in scientific publishing. Preprint outcomes reflect particularly large differences between countries: comparatively wealthy countries in Europe and North America post more preprints, which are downloaded more frequently, published more consistently, and favored by the largest and most well-known journals in biology. While there are many potential explanations for these dynamics, the quantification of these patterns may help stakeholders make more informed decisions about how they read, write and publish preprints in the future.

## Methods

### Ethical statement

This study was submitted to the University of Minnesota Institutional Review Board (study #00008793), which determined the work did not qualify as human subjects research and did not require IRB oversight.

### Preprint metadata

We used existing data from the Rxivist web crawler (*Abdill and Blekhman, 2019c*) to build a list of URLs for every preprint on bioRxiv.org. We then used this list as the input for a new tool that collects author data: we recorded a separate entry for each author of each preprint, and stored name, email address, affiliation, ORCID identifier, and the date of the most recent version of the preprint that has been indexed in the Rxivist database. While the original web crawler performs author consolidation during the paper index process (i.e. 'Does this new paper have any authors we already recognize?'), this new tool creates a new entry for each preprint; we make no connections for authors across preprints in this analysis, and infer author country separately for every author of every paper. It is also important to note that for longitudinal analyses of preprint trends, each preprint is associated with the date on *its most recent version*, which means a paper first posted in 2015, but then revised in 2017, would be listed in 2017. The final version of the preprint metadata was collected in the final weeks of January 2020—because preprints were filtered using the most recent known date, those posted before 2020, but revised in the first month of 2020, were not included in the analysis. In addition, 95 preprints were excluded because the bioRxiv website repeatedly returned errors when we tried to collect the metadata, leaving a total of 67,885 preprints in the analysis. Of these, there were 2409 manuscripts (3.6%) for which we were unable to scrape affiliation data for at least one author, including 137 preprints with no affiliation information for any author.

bioRxiv maintains an application programmatic interface (API) that provides machine-readable data about their holdings. However, the information it exposes about authors and their affiliations is not as complete as the information available from the website itself, and only the corresponding author's institutional affiliation is included (https://api.biorxiv.org/). Therefore, we used the more complete data in the Rxivist database (*Abdill and Blekhman, 2019b*), which includes affiliations for all authors.

All data on published preprints was pulled directly from bioRxiv. However, it is also possible, if not likely, that the publication of many preprints goes undetected by its system. *Fraser et al., 2020* developed a method of searching for published preprints in Scopus and Crossref databases and found most had already been picked up by bioRxiv's detection process, though bioRxiv states that preprints published with new titles or authors can go undetected (https://www.biorxiv.org/about-biorxiv), and preliminary data suggests this may affect thousands of preprints (*Abdill and Blekhman, 2019b*). How these effects differ by country of origin

remains unclear—perhaps authors from some countries are more likely to have their titles changed by journal editors, for example—but bias at the country level may also be more pronounced for other reasons. The assignment of Digital Object Identifiers (DOIs) to papers provides a useful proxy for participation in the 'western' publishing system. Each published bioRxiv preprint is listed with the DOI of its published version, but DOI assignment is not yet universally adopted. *Boudry and Chartron, 2017* examined papers from 2015 indexed by PubMed and found DOI assignment varied widely based on the country of the publisher. 96% of publications in Germany had a DOI, for example, plus 98% of UK publications and more than 99% of Brazilian publications. However, only 31% of papers published in China had DOIs, and just 2% (33 out of 1582) of papers published in Russia. There are 45 countries that overlap between our analysis and that of *Boudry and Chartron, 2017*. Of these, we found a modest correlation (Spearman's rho = 0.295, p=0.0489) between a country's preprint publication rate and the rate at which publishers in that country assigned DOIs (*Figure 4—source data 2*). This indicates that countries with higher rates of DOI issuance (for publications dating back to 1955) also tend to have higher observed rates of preprint publication.

### Attribution of preprints

Throughout the analysis, we define the 'senior author' for each preprint as the author appearing last in the author list, a longstanding practice in biomedical literature (*Riesenberg, 1990*; *Buehring et al., 2007*) corroborated by a 2003 study, which found that 91% of publications indicated a corresponding author that was in the first- or last-author position (*Mattsson et al., 2011*). Among the 59,562 preprints for which the country was known for the first and last author, 7965 (13.4%) preprints included a first author associated with a different country than the senior author.

When examining international collaboration, we also considered whether more nuanced methods of distributing credit would be more informative. Our primary approach—assigning each preprint to the one country appearing in the senior author position—is considered *straight counting* (*Gauffriau et al., 2008*). We repeated the process using *complete-normalized counting* (*Figure 1—source data 2*), which splits a single credit among all authors of a preprint. So, for a preprint with 10 authors, if six

authors are affiliated with an institution in the UK, the UK would receive 0.6 'credits' for that preprint. We found the complete-normalized preprint counts to be almost identical to the counts distributed based on straight counting (Pearson's r = 0.9971, p=4.48 $\times$ $10^{-197}$). While there are numerous proposals for proportioning differing levels of recognition to authors at different positions in the author list (e.g. *Hagen, 2013*; *Kim and Diesner, 2015*), the close link between the complete-normalized count and the count based on senior authorship indicates that senior authors are at least an accurate proxy for the overall number of individual authors, at the country level.

When computing the average authors per paper, the harmonic mean is used to capture the average 'contribution' of an author, as in *Glänzel and Schubert, 2005*—in short, this shows that authors were responsible for about one-third of a preprint in 2014, but less than one-fourth of a preprint as of 2019.

### Data collection and management

All bioRxiv metadata was collected in a relational PostgreSQL database (https://www.postgresql.org). The main table, 'article_authors,' recorded one entry for each author of each preprint, with the author-level metadata described above. Another table associated each unique affiliation string with an inferred institution (see **Institutional affiliation assignment** below), with other tables linking institutions to countries and preprints to publications. (See the repository storing the database snapshot for a full description of the database schema.) Analysis was performed by querying the database for different combinations of data and outputting them into CSV files for analysis in R (*R Core Team, 2019*). For example, data on 'authors per preprint' was collected by associating all the unique preprints in the 'article_authors' table with a count of the number of entries in the table for that preprint. Similar consolidation was done at many other levels as well—for example, since each author is associated with an affiliation string, and each affiliation string is associated with an institution, and each institution is associated with a country, we can build queries to evaluate properties of preprints grouped by country.

### Contributor countries

The analysis described in the 'Collaboration' section measured correlations between three country-level descriptors, calculated for all countries

that contributed to more than 50 international preprints:

i. International collaborations. The total number of international preprints including at least one author from that country.
ii. International collaboration rate. Of all preprints listing an author from that country, the proportion of them that includes at least one author from another country.
iii. International senior-author rate. Of all the international collaborations associated with a country, the proportion of them for which that country was listed as the senior author.

We examined disproportionate links between contributor countries and senior-author countries by performing one-tailed Fisher's exact tests between each contributor country and each senior-author country, to test the null hypothesis that there is no association between the classifications 'preprints with an author from the contributor country' and preprints with a senior author from the senior-author country.' To minimize the effect of partnerships between individual researchers affecting country-level analysis, the senior-author country list included only countries with at least 25 senior-author preprints that include a contributor country, and we only evaluated links between contributor countries and senior-author countries that included at least five preprints. We determined significance by adjusting p-values using the Benjamini–Hochberg procedure.

### BioRxiv adoption

When evaluating bioRxiv participation, we corrected for overall research output, as documented by SCImago Journal and Country Rank portal, which counts articles, conference papers, and reviews in Scopus-indexed journals (https://www.scimagojr.com; https://www.scimagojr.com/help.php). We added the totals of these 'citable documents' from 2014 through 2019 for each countries with at least 3000 citable documents and 50 preprints. We used these totals to generate a productivity-adjusted score, termed 'bioRxiv adoption,' by taking the proportion of preprints with a senior author from that country and dividing it by that country's proportion of citable documents from 2014 to 2019. While SCImago is not specific to life sciences research, it was chosen over alternatives because it had consistent data for all countries in our dataset. A shortcoming of combining data SCImago and

the Research Organization Registry (see below) is that they use different criteria for the inclusion of separate states. In most cases, SCImago provides more specific distinctions than ROR: for example, Puerto Rico is listed separately from the US in the SCImago dataset, but not in the ROR dataset. We did not alter these distinctions—as a result, nations with disputed or complex borders may have slightly inflated bioRxiv adoption scores. For example, preprints attributed to institutions in Hong Kong are counted in the total for China, but the 108,197 citable documents from Hong Kong in the SCImago dataset are not included in the China total.

### Visualization

All figures were made with R and the ggplot2 package (*Wickham, 2016*), with colors from the RcolorBrewer package (*Neuwirth, 2014*; *Woodruff and Brewer, 2017*). World maps were generated using the Equal Earth projection (*Šavrič et al., 2019*) and the rnaturalearth R package (*South, 2017*), following the procedure described in *Le et al., 2020*. Code to reproduce all figures is available on GitHub (*Abdill, 2020*; https://github.com/blekhmanlab/biorxiv_countries; copy archived at https://github.com/elifesciences-publications/biorxiv_countries).

### Institutional affiliation assignment

We used the Research Organization Registry (ROR) API to translate bioRxiv affiliation strings into canonical institution identities (*Research Organization Registry, 2019*). We launched a local copy of the database using their included Docker configuration and linked it to our web crawler's container, to allow the two applications to communicate. We then pulled a list of every unique affiliation string observed on bioRxiv and submitted them to the ROR API. We used the response's 'chosen' field, indicating the ROR application's confidence in the assignment, to dictate whether the assignment was recorded. Any affiliation strings that did not have an assigned result were put into a separate 'unknown' category. As with any study of this kind, we are limited by the quality of available metadata. Though we are able to efficiently scrape data from bioRxiv, data provided by authors can be unreliable or ambiguous. There are 465 preprints, for example, in which multiple or all authors on a paper are listed with the same ORCID, ostensibly a unique personal identifier, and there are hundreds of preprints for

which authors do not specify any affiliation information at all, including in the PDF manuscript itself. We are also limited by the content of the ROR system (https://ror.org/about/): Though there are tens of thousands of institutions in the dataset and its basis, the Global Research Identifier Database (https://www.grid.ac/stats), has extensive coverage around the world, the translation of affiliation strings is likely more effective for regions that have more extensive coverage.

### Country level accuracy of ROR assignments

Across 67,885 total preprints, we indexed 488,660 total author entries, one for each author of each preprint. These entries each included one of 136,456 distinct affiliation strings, which we processed using the ROR API before making manual corrections.

We first focused on assigning countries to preprints that were in the 'unknown' category. We started by manually adding institutional assignments to 'unknown' affiliation strings that were associated with 10 or more authors. We then used sub-strings within affiliation strings to find matches to existing institutions, and finally generated a list of individual words that appeared most frequently in 'unknown' affiliation strings. We searched this list for words indicating an affiliation that was at least as specific as a country (e.g. 'Italian,' 'Boston,' 'Guangdong') and associated any affiliation strings that included that word with an institution in the corresponding country. Finally, we evaluated any authors still in the 'unknown' category by searching for the presence of a country-specific top-level domain in their email addresses—for example, uncategorized authors with an email address ending in '.nl' were assigned to the Netherlands. Generic domains such as '.com' were not categorized, with the exception of '.edu,' which was assigned to the US. While these corrections would have negatively impacted the institution-level accuracy, it was a more practical approach to generate country-level observations.

There were also corrections made that placed *more* affiliations into the 'unknown' category—there is an ROR institution called 'Computer Science Department,' for example, that contained spurious assignments. Prior to correction, 23,158 (17%) distinct affiliation strings were categorized as 'unknown,' associated with 71,947 authors. After manual corrections, there were 20,099 unknown affiliation strings associated with 51,855 authors.

There were also corrections made to existing institutional assignments, which are used to make the country-level inferences about author location. It appears the ROR API struggles with institutions that are commonly expressed as acronyms—affiliation strings including 'MIT,' for example, was sometimes incorrectly coded not as 'Massachusetts Institute of Technology' in the US, but as 'Manukau Institute of Technology' in New Zealand, even when other clues within the affiliation string indicated it was the former. Other affiliation strings were more broadly opaque— 'Centre for Research in Agricultural Genomics (CRAG) CSIC-IRTA-UAB-UB,' for example. A full list of manual edits is included in the 'manual_edits.sql' file.

In total, 12,487 institutional assignments were corrected, affecting 52,037 author entries (10.6%). Prior to the corrections, an evaluation of the ROR assignments in a random sample (n = 488) found the country-level accuracy was 92.2 ± 2.4%, at a 95% confidence interval. After an initial round of corrections, the country-level accuracy improved to 96.5 ± 1.6%. (These samples were sized to evaluate errors in the assignment of institutions rather than countries, which, once institution-level analysis was removed from the study, became irrelevant.) After another round of corrections that assigned countries to 14,690 authors in the 'unknown' category, we pulled another random sample of corrected affiliations—using a 95% confidence interval, the sample size required to detect 96.5% assignment accuracy with a 2% margin of error was calculated to be 325 (*Naing et al., 2006*). Manually evaluating the country assignments of this sample showed the country-level accuracy of the corrected affiliations was 95.7 ± 2.2%.

Though preprints that were assigned a country could be categorized with high accuracy, we also sought to characterize the preprints that remained in the 'unknown' category after corrections, to evaluate whether there was a bias in which preprints were categorized at all. Among the successfully classified preprints, the distribution across countries is heavily skewed—the 27 most prolific countries (15%) account for 95.3% of categorized preprints. Accordingly, characterizing the prevalence of individual countries would require an impractically large sample made up of a large portion of all uncategorized preprints. Instead, we split the countries into two groups: the first contained the 27 most prolific countries. The second group contained the remaining 148 countries, which account for the remaining 2960 preprints (4.7%). We used this

as the prevalence in our sample size calculation. Using a 95% confidence interval and a precision of 0.00235 (half the prevalence), the sample size (with correction for a finite population of 4,985) was calculated to be 307 (*Naing et al., 2006*). Within this sample, we found that preprints with a senior author in the bottom 148 countries were present at a prevalence of 12.6 ± 3.9%.

## Acknowledgements
We thank Alex D Wade (Chan Zuckerberg Initiative) for his insights on author disambiguation and the members of the Blekhman lab for helpful discussions. We also thank the Research Organization Registry community for curating an extensive, freely available dataset on research institutions around the world.

**Richard J Abdill** is in the Department of Genetics, Cell Biology, and Development, University of Minnesota, Minneapolis, United States

iD https://orcid.org/0000-0001-9565-5832

**Elizabeth M Adamowicz** is in the Department of Genetics, Cell Biology, and Development, University of Minnesota, Minneapolis, United States

iD https://orcid.org/0000-0003-3332-8021

**Ran Blekhman** is in the Department of Genetics, Cell Biology, and Development and the Department of Ecology, Evolution and Behavior, University of Minnesota, Minneapolis, United States

blekhman@umn.edu

iD https://orcid.org/0000-0003-3218-613X

*Author contributions:* Richard J Abdill, Conceptualization, Data curation, Software, Formal analysis, Visualization, Methodology, Writing - original draft; Elizabeth M Adamowicz, Data curation, Methodology, Writing - original draft; Ran Blekhman, Conceptualization, Supervision, Funding acquisition, Methodology, Writing - review and editing

*Competing interests:* Richard J Abdill: Has been a volunteer ambassador for ASAPbio, an open-science advocacy organization that is also affiliated with Review Commons. The other authors declare that no competing interests exist.

## Funding

| Funder | Grant reference number | Author |
|---|---|---|
| National Institutes of Health | R35-GM128716 | Ran Blekhman |
| University of Minnesota | McKnight Land-Grant Professorship | Ran Blekhman |

The funders had no role in study design, data collection and interpretation, or the decision to submit the work for publication.

**Decision letter and Author response**
Decision letter https://doi.org/10.7554/eLife.58496.sa1
Author response https://doi.org/10.7554/eLife.58496.sa2

## Additional files
### Supplementary files
• Transparent reporting form

### Data availability
All data has been deposited in a versioned repository at Zenodo.org. Source data files have been provided for all figures, along with the code used to generate each plot. Code used to collect and analyze data has been deposited at https://github.com/blekhmanlab/biorxiv_countries (copy archived at https://github.com/elifesciences-publications/biorxiv_countries).

The following dataset was generated:

| Author(s) | Year | Dataset URL | Database and Identifier |
|---|---|---|---|
| Abdill RJ, Adamowicz EM, Blekhman R | 2020 | https://doi.org/10.5281/zenodo.3762814 | Zenodo, 10.5281/zenodo.3762814 |

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
