## [Decision Letter]

Thank you for submitting your article "Meta-research: International authorship and collaboration across all bioRxiv preprints" to *eLife*. I recruited two reviewers for your manuscript but, unfortunately, one of them had to drop out. However, on the basis of the report from the other reviewer (Jacob J Hughey; see below), I would like to invite you to submit a revised version of your manuscript that addresses the points below.

Summary:

Abdill et al. parsed author affiliations of bioRxiv preprints to explore country-wise differences in preprint submissions, international collaborations, and publication rates. Most of the work appears technically solid. The results are interesting but could be presented more clearly and precisely. In my opinion, some of the interpretations need more evidence behind them or need to be toned down. I would also encourage the authors to consider what analyses might make the story more compelling.

Essential revisions:

1) The "Unknown" country affiliations account for more preprints than every country other than the United States. Although the authors made an effort to reduce the number of unknown affiliations, the unknowns make it difficult to draw conclusions about countries with small numbers of preprints or any long term trends. I would suggest trying to reduce the number of unknowns or show they're not missing from specific regions (e.g., are certain languages over-represented in institution names amongst unknown affiliations?) and/or restating results to acknowledge the uncertainty.

2) For related plots and analysis, the authors used different cut-off values without any clear justification as to why. For example, in defining contributors, the authors require at least 50 international preprints, but in the subsequent analysis, this cut-off is changed to countries with at least 30 international preprints to add additional countries. Similarly, Figure 5a uses 15 preprints for looking at journal-country links but uses 30 preprints for looking at the US specifically.

3) The characterization of contributor countries and their publication statistics is not entirely convincing. I could imagine identifying similar countries by clustering, as opposed to using arbitrary cut-offs. I actually find Figure 3-figure supplement 1 more informative than Figure 3 itself. I don't think the map adds much, and one could add country names to the scatterplots. On a related note, these appear to be adjusted p-values based on the methods, but for clarity, this should be stated in the text and/or caption.

- Note from the eLife Features Editor: Please change Figure 3a into a figure supplement to Figure 3, but please leave the present Figure 3-figure supplement 1 as a figure supplement.

4) It's unclear to me exactly how the authors calculated the number of downloads for each preprint. Did they calculate the mean or median number of downloads per month for each preprint prior to publication, then the median (of the mean or median) for preprints from each country? If instead they just used the total number of downloads for each preprint, that would seem to be confounded by when the preprint was posted (perhaps benefitting early adopter countries) and when it was published (perhaps benefitting countries with lower publication rates).

5) Throughout the manuscript, but especially in the analysis of associations between countries and journals, I found myself wondering about country-level differences in preprint submissions in different subject areas. Granted, many of the journals with overrepresentation of U.S. research are general (e.g., Cell, Science, PNAS), but still I wonder about uptake of preprints in different communities.

- Note from the eLife Features Editor: You do not need to add any data or figures in response to this point, but please discuss it in the text.

[Editors' note: further revisions were requested prior to acceptance, as described below.]

Thank you for submitting the revised version of your article, "Meta-research: International authorship and collaboration across bioRxiv preprints". I sent the revised version to the reviewer who reviewed the original version, and I am pleased to say that they support publication, though they have raised a small number of points that you might want to address (see below).

While an interesting result, it remains unclear to me what the download and publication rate correlation is supposed to convey. The authors ask some open-ended questions in the Discussion regarding this finding, but these don't strike me as ones best analyzed from the perspective of preprints.

In the discussion of low bioRxiv adoption, one consideration might be the availability of domestic preprint servers. For example, China, one of the low-adoption countries mentioned in the results, has internal domestic preprint servers that may be preferred over bioRxiv.

---

## [Author Response]

[We repeat the reviewers’ points here in italic, and include our responses point by point, as well as a description of the changes made, in Roman.]

Essential revisions:1) The "Unknown" country affiliations account for more preprints than every country other than the United States. Although the authors made an effort to reduce the number of unknown affiliations, the unknowns make it difficult to draw conclusions about countries with small numbers of preprints or any long term trends. I would suggest trying to reduce the number of unknowns or show they're not missing from specific regions (e.g., are certain languages over-represented in institution names amongst unknown affiliations?) and/or restating results to acknowledge the uncertainty.

We thank the reviewer for this helpful observation. We’ve made further corrections to affiliation strings categorized as “unknown,” assigning a country to an additional 14,690 authors (a 22% improvement) and reducing the number of papers with an “unknown” country association from 9,077 to 4,985, a 45% improvement. We used these new corrected assignments to recalculate all results reported in the paper, and generate updated versions of all figures. In addition, we re-evaluated the accuracy of the assignments and added a description to the Methods section:

"In total, 12,487 institutional assignments were corrected, affecting 52,037 author entries (10.6%). Prior to the corrections, an evaluation of the ROR assignments in a random sample (n=488) found the country-level accuracy was 92.2% ± 2.4%, at a 95% confidence interval. After an initial round of corrections, the country-level accuracy improved to 96.5% ± 1.6%. (These samples were sized to evaluate errors in the assignment of institutions rather than countries, which, once institution-level analysis was removed from the study, became irrelevant.) After another round of corrections that assigned countries to 14,690 authors in the "unknown" category, we pulled another random sample of corrected affiliations—using a 95% confidence interval, the sample size required to detect 96.5% assignment accuracy with a 2% margin of error was calculated to be 325 (Naing, Winn & Rusli 2006). Manually evaluating the country assignments of this sample showed the country-level accuracy of the corrected affiliations was 95.7% ± 2.2%."

We also added a new analysis to characterize the preprints found in the “unknown” category after correction, to help determine whether some countries are more likely to be missed by the classification system, which we added to the Methods as well:

"Though preprints that were assigned a country could be categorized with high accuracy, we also sought to characterize the preprints that remained in the "unknown" category after corrections, to evaluate whether there was a bias in which preprints were categorized at all. Among the successfully classified preprints, the distribution across countries is heavily skewed: The mean preprints per country is 359, or 0.57% of the total. Accordingly, characterizing the prevalence of individual countries would require an impractically large sample made up of a large portion of all uncategorized preprints. Instead, we split the countries into two groups: the first contained the 27 most prolific countries, which account for 95.3% of categorized preprints. The second group contained the remaining 148 countries. These account for the remaining 2,960 preprints (4.7%), which we used as the prevalence in our sample size calculation. Using a 95% confidence interval and a precision of 0.00235 (half the prevalence), the sample size (with correction for a finite population of 4,985) was calculated to be 307 (Naing, Winn & Rusli 2006). Within this sample, we found that preprints with a senior author in the bottom 148 countries were present at a prevalence of 12.6% ± 3.9%."

We added a brief evaluation of this limitation to the Discussion section:

"There were also 4,985 preprints (7.3%) for which we were not able to confidently assign a country of origin. An evaluation of these (see Methods) showed that the most prolific countries were also underrepresented in the "unknown" category, compared to the 148 other countries with at least one author. While it is impractical to draw country-specific conclusions from this, it suggests that the preprint counts for countries with comparatively low participation may be slightly higher than reported, an issue that may be exacerbated in more granular analyses, such as at the institutional level."

We also added a new figure (Figure 1–figure supplement 2) to summarize these findings.

2) For related plots and analysis, the authors used different cut-off values without any clear justification as to why. For example, in defining contributors, the authors require at least 50 international preprints, but in the subsequent analysis, this cut-off is changed to countries with at least 30 international preprints to add additional countries. Similarly, Figure 5a uses 15 preprints for looking at journal-country links but uses 30 preprints for looking at the US specifically.

We agree with this comment, and have revised the analysis of contributor countries to use the same 50-preprint cutoff for all analyses. The text has been updated to reflect the new thresholds, and the relevant figure supplement (now Figure 3–figure supplement 2) has been redrawn to show the new subset of countries used in the analysis.

The other inconsistent thresholds for journal–country links have also been revised so that all analyses described in Figure 5 use the same cutoff of 15 preprints. Figure 5 has also been revised.

3) The characterization of contributor countries and their publication statistics is not entirely convincing. I could imagine identifying similar countries by clustering, as opposed to using arbitrary cut-offs. I actually find Figure 3-figure supplement 1 more informative than Figure 3 itself. I don't think the map adds much, and one could add country names to the scatterplots. On a related note, these appear to be adjusted p-values based on the methods, but for clarity, this should be stated in the text and/or caption.- Note from the eLife Features Editor: Please change Figure 3a into a figure supplement to Figure 3, but please leave the present Figure 3-figure supplement 1 as a figure supplement.

The previous Figure 3a has been moved to a figure supplement (now Figure 3–figure supplement 1), and Figure 3 has been re-drawn to use fewer panels. We also added a sentence to the Figure 3 caption reflecting the multiple test correction:

"All p-values reflect multiple-test correction using the Benjamini–Hochberg procedure."

Regarding the definition of “contributor countries,” we agree that a clustering approach may provide more interpretable boundaries. We were not able to build a model of collaboration that adequately addressed the important question of why some countries appear on preprints in consistently subordinate roles. Using the three measurements visualized in Figure 3–figure supplement 1 (now supplement 2), we have added a new, interactive visualization that emphasizes the point that the cutoff itself is relatively unimportant because all countries in the analysis fall inside a constrained space—defining a concrete notion of “contributor” is used only to highlight the properties of one end of this space. The new Figure 3–Figure supplement 3 illustrates the highly correlated measurements along the three axes in question.

4) It's unclear to me exactly how the authors calculated the number of downloads for each preprint. Did they calculate the mean or median number of downloads per month for each preprint prior to publication, then the median (of the mean or median) for preprints from each country? If instead they just used the total number of downloads for each preprint, that would seem to be confounded by when the preprint was posted (perhaps benefitting early adopter countries) and when it was published (perhaps benefitting countries with lower publication rates).

We initially used total downloads per preprint in our calculations, and we thank the reviewer for the important observation that this could introduce bias against countries that have adopted bioRxiv usage more recently. To avoid this potential bias, we have revised all calculations to use only the first six months of download counts for each preprint and clarified this calculation in the Results:

"When evaluating downloads per preprint, we used only download numbers from each preprint's first six months online, which would capture the majority of downloads for most preprints (Abdill & Blekhman 2019b) while minimizing the effect of the "long tail" of downloads that would be longer for countries that were earlier adopters. Using this measurement, the median number of PDF downloads per preprint is 210 (Figure 4a)."

5) Throughout the manuscript, but especially in the analysis of associations between countries and journals, I found myself wondering about country-level differences in preprint submissions in different subject areas. Granted, many of the journals with overrepresentation of U.S. research are general (e.g., Cell, Science, PNAS), but still I wonder about uptake of preprints in different communities.- Note from the eLife Features Editor: You do not need to add any data or figures in response to this point, but please discuss it in the text.

This is an interesting point that could merit further research. Given the instruction from the editor, we have added several sentences to the “limitations” section of the Discussion to reflect this:

"Country-level differences in metrics such as downloads and publication rate may also be confounded with field-level differences: On average, genomics preprints are downloaded twice as many times as microbiology preprints (Abdill & Blekhman 2019b), for example, so countries with a disproportionate number of preprints in a particular field could receive more downloads due to choice of topic, rather than country of origin. Further study is required to determine whether these two factors are related and in which direction."

[Editors' note: further revisions were requested prior to acceptance, as described below.]

While an interesting result, it remains unclear to me what the download and publication rate correlation is supposed to convey. The authors ask some open-ended questions in the Discussion regarding this finding, but these don't strike me as ones best analyzed from the perspective of preprints.

We have added a sentence (shown in italic below) to this section in the Discussion to clarify the importance of this finding:

Second, we found numerous country-level differences in preprint outcomes, *including a positive correlation at the country level between downloads per preprint and publication rate (Figure 4c). This raises an important consideration that when evaluating the role of preprints, some benefits may be realized by authors in some countries more consistently than others*. If one of the goals of preprinting one’s work is to solicit feedback from the community (Sarabipour et al. 2019; Sever et al. 2019), what are the implications of the average Brazilian preprint receiving 37% fewer downloads than the average Dutch preprint?

In the discussion of low bioRxiv adoption, one consideration might be the availability of domestic preprint servers. For example, China, one of the low-adoption countries mentioned in the results, has internal domestic preprint servers that may be preferred over bioRxiv.

We have added a caveat (shown in italic below) to the Discussion section to reflect other places preprints may be found:

Open science advocates have argued that there should not be a "one size fits all" approach to preprints and open access (Debat and Babini 2020; ALLEA 2018; Becerril-García 2019; Mukunth 2019), and further research is required to determine what drives certain countries to preprint servers—what incentives are present for biologists in Finland but not Greece, for example. *There is also more to be done regarding the trade-offs of using a more distributed set of repositories that are specific to disciplines or countries (e.g. INA-Rxiv in Indonesia or PaleorXiv for paleontology), which could also influence the observed levels of bioRxiv adoption.* Our results make it clear that those reading bioRxiv (or soliciting submissions from the platform) are reviewing a biased sample of worldwide scholarship.